# *Dactylonectria* and *Ilyonectria* Species Causing Black Foot Disease of Andean Blackberry (*Rubus Glaucus* Benth) in Ecuador

Jessica Sánchez [1], Paola Iturralde [1], Alma Koch [1], Cristina Tello [2], Dennis Martinez [1], Natasha Proaño [1], Anibal Martínez [3], William Viera [3], Ligia Ayala [1] and Francisco Flores [1,4,*]

[1] Departamento de Ciencias de la Vida y la Agricultura, Universidad de las Fuerzas Armadas-ESPE, Av. General Rumiñahui, Sangolquí 171103, Ecuador; jvsanchez1107@gmail.com (J.S.); pjiturralde@espe.edu.ec (P.I.); arkoch@espe.edu.ec (A.K.); dnismartinezm@gmail.com (D.M.); tasha_2011@hotmail.com (N.P.); liayala@espe.edu.ec (L.A.)
[2] Plant Protection Department, National Institute of Agricultural Research (INIAP), Panamericana Sur km 1, Cutuglahua 171108, Ecuador; cristina.tello@iniap.gob.ec
[3] Fruit Program, National Institute of Agricultural Research (INIAP), Av. Interoceánica km 14 ½, Tumbaco 170184, Ecuador; anibal.martinez@iniap.gob.ec (A.M.); william.viera@iniap.gob.ec (W.V.)
[4] Centro de Investigación de Alimentos, CIAL, Facultad de Ciencias de la Ingeniería e Industrias, Universidad UTE, Quito 170147, Ecuador
* Correspondence: fjflores2@espe.edu.ec

**Abstract:** Andean blackberry (*Rubus glaucus* Benth) plants from the provinces of Tungurahua and Bolivar (Ecuador) started showing symptoms of black foot disease since 2010. Wilted plants were sampled in both provinces from 2014 to 2017, and fungal isolates were obtained from tissues surrounding necrotic lesions in the cortex of the roots and crown. Based on morphological characteristics and DNA sequencing of histone 3 and the translation elongation factor 1α gene, isolates were identified as one of seven species, *Ilyonectria vredehoekensis*, *Ilyonectria robusta*, *Ilyonectria venezuelensis*, *Ilyonectria europaea*, *Dactylonectria torresensis*, or *Dactylonectria novozelandica*. Pathogenicity tests with isolates from each species, excluding *I. europaea* and *D. novozelandica* whose isolates were lost due to contamination, confirmed that the four species tested can produce black foot disease symptoms in Andean blackberry. This is the first report of *Dactylonectria* and *Ilyonectria* species causing black foot disease of Andean blackberry.

**Keywords:** *Cylindrocarpon*-like anamorphs; root rot; soilborne pathogens; wilt

## 1. Introduction

Species with *Cylindrocarpon*-like anamorphs include several taxa that are common soilborne plant pathogens; these fungi usually form chlamydospores that allow them to survive for long dormancy periods and have a wide host range that includes woody and herbaceous plants in which they cause severe damage [1,2]. Among this group of fungi, *Ilyonectria* and *Dactylonectria* species are described as the causal agents of black foot of grapevines (*Vitis* spp.) [1,3,4], a disease that has been extensively studied since 1961 when it was first described in France [5]. *Cylindrocarpon*-like anamorphs have also been associated with black root rot of strawberries (*Fragaria* × *ananassa* Duch.) and raspberries (*Rubus* sp.) [6], and black foot of Andean blackberry (*Rubus glaucus* Benth) [7].

In the state of Mérida, Venezuela, black foot disease of Andean blackberry was associated with *Cylindrocarpon destructans* (Zinssm.) Scholten, recognized at the time as a species complex, and *Cylindrocarpon ianthothele* Wollenw., in 2004 [7]. Since then, taxonomic aspects of *Cylindrocarpon*-like

anamorphs have been clarified through multilocus phylogenetic analyses, which granted the description of new genera, including *Dactylonectria* and *Ilyonectria*, and new species within these genera [8–10]. Studies based on the analysis of ribosomal DNA sequences showed that that *Neonectria/Cylindrocarpon* genus was formed by three distinct monophyletic groups [11]. Years later, Chaverri et al. [1] used the phylogenetic signal of six genes (*act*, ITS, LSU, *rpb1*, *tef1*, *tub*) to divide *Neonectria* into five new genera, *Neonectria sensu stricto*, *Rugonectria*, *Thelonectria*, *Ilyonectria*, and *Campylocarpon*. *Neonectria radicicola*, the teleomorph of the *C. destructans*, was then assigned as a type of *Ilyonectria* (*I. radicicola*) and later renamed as *Ilyonectria destructans* (Zinssm.) Rossman, L. Lombard, and Crous [12]. Subsequent studies divided the *I. destructans* species complex into several new species [8], including causal agents of black foot disease of grapevine [10]. The genus *Dactylonectria* with 10 new combinations, which were mostly previously treated as *Ilyonectria*, was introduced in 2014 by Lombard et al. [13]. The lack of molecular information on the isolates described in the first report of black foot disease of Andean blackberry makes it impossible to confirm their classification under the current taxonomy.

Symptoms associated with black foot disease were observed in Andean blackberry plantations in Ecuador since 2010. During 2017, in the provinces of Tungurahua and Bolivar black foot had an incidence of 13.3%, generating significant economic losses to blackberry farmers. The disease is characterized by wilting of the plant, necrotic lesions expanding through the cortex and the vasculature of the roots and the crown, foliage chlorosis, and curling of the leaf tips. The damage suffered in the roots and crown, prevents sprouting and hinders development of plants, resulting in decreased productivity. The objective of this study was to identify the causal agent of black foot disease of Andean blackberry in Ecuador, using morphological characteristics, molecular methods, and pathogenicity tests.

## 2. Materials and Methods

### 2.1. Isolates from Symptomatic Plants

Thirty Andean blackberry plants showing symptoms of black foot (Figure 1) were collected in different locations from the provinces of Tungurahua and Bolivar from 2014 to 2017. Plant roots and crowns were washed with tap water to remove soil. Small segments (5 mm) were taken from the tissue surrounding lesions, surface sterilized with a 0.5% NaClO solution for 3 min, rinsed three times with sterile distilled water, placed in Petri dishes containing potato dextrose agar (PDA) with chloramphenicol (100 ppm), and incubated for seven days at room temperature (20 ± 2 °C). Single-spore isolates were obtained from the fungal colonies growing on the media.

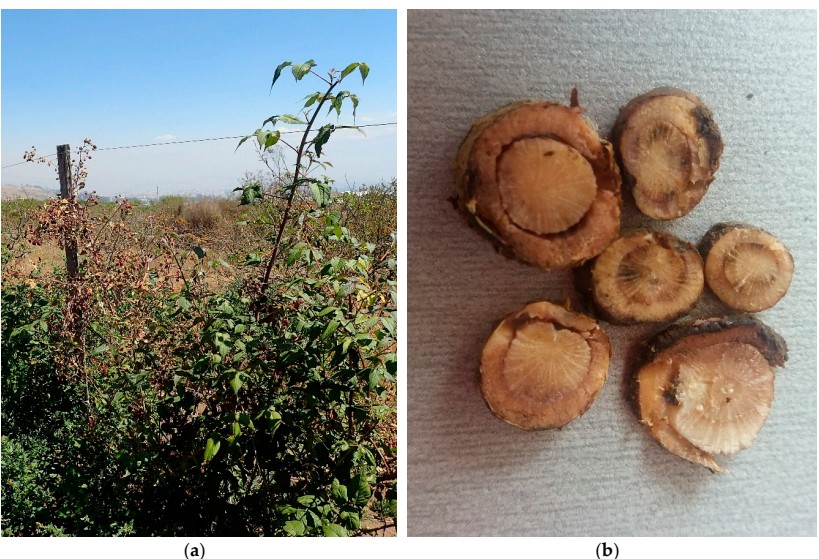

(a)                                                            (b)

**Figure 1.** Symptoms of black foot in plants of Andean blackberry. (**a**) Wilted plant next to a healthy one and (**b**) cross sections of the crown of diseased plants showing necrosis of vascular bundles.

## 2.2. Morphological Characterization

Macroscopic characteristics were determined for isolates growing on PDA with chloramphenicol (100 ppm) for 4 weeks at 25 °C. For the microscopic characterization, we used a microculture slide that contained of a 3 × 2 mm piece of PDA placed next to an equivalent piece of PDA with mycelium that served as inoculum, with a coverslip on top. The slide was placed inside a Petri plate with filter paper soaked with 4 mL of sterile distilled water using toothpicks to separate the slide from the filter paper. The Petri plate was sealed with parafilm and incubated at 25 °C for seven days. From each isolate, 30 conidia were measured using the cellSens Standard software (Olympus). Images were captured with the Olympus DP73 optical microscope.

## 2.3. Molecular Identification and Phylogenetic Analysis

Seventeen isolates (Table 1) were cultivated in potato dextrose broth (PDB), at 80 rpm for 10 days at room temperature. Mycelia were filtered and freeze dried, and the genomic DNA was extracted using the PureLink genomic plant DNA Purification Kit (Invitrogen), following the manufacturer's instructions.

**Table 1.** *Ilyonectria* and *Dactylonectria* translation elongation factor 1 alpha (*TEF*) and histidine 3 (*HIS3*) sequences.

| Isolate | Species | Year | Province/Location | Accession Number | |
|---|---|---|---|---|---|
| | | | | TEF | HIS3 |
| CG2 | *Ilyonectria venezuelensis* | 2014 | Bolivar/Guaranda | MG852016 | MG852002 |
| CG3 | *Ilyonectria europaea* | 2014 | Bolivar/Guaranda | MG852017 | MG852003 |
| CG4 | *Ilyonectria vredehoekensis* | 2014 | Bolivar/Guaranda | - | MG852004 |
| CG5 | *I. vredehoekensis* | 2014 | Bolivar/Guaranda | - | MG852005 |
| C4 | *Dactylonectria torresensis* | 2014 | Tungurahua/Ambato | MG852018 | MG851999 |
| C6 | *I. vredehoekensis* | 2014 | Tungurahua/Ambato | - | MG852000 |
| C7 | *Dactylonectria novozelandica* | 2014 | Tungurahua/Ambato | MG852019 | MG852001 |
| UFAH00033 | *D. torresensis* | 2016 | Tungurahua/Tisaleo | KY114517 | MG852015 |
| UFAH00034 | *I. vredehoekensis* | 2017 | Bolivar/Chillanes | MG852020 | MG852014 |
| UFAH00035 | *I. vredehoekensis* | 2017 | Bolivar/Chillanes | MG852021 | MG852006 |
| UFAH00036 | *I. vredehoekensis* | 2017 | Bolivar/Chillanes | MG852022 | MG852013 |
| UFAH00037 | *I. vredehoekensis* | 2017 | Tungurahua/Píllaro | - | MG852010 |
| UFAH00038 | *Ilyonectria* sp. | 2017 | Tungurahua/Ambato | - | MG852009 |
| UFAH00039 | *I. venezuelensis* | 2017 | Tungurahua/Tisaleo | MG852023 | MG852012 |
| UFAH00040 | *I. vredehoekensis* | 2017 | Bolivar/Chillanes | - | MG852008 |
| UFAH00041 | *I. venezuelensis* | 2017 | Tungurahua/Ambato | MG852024 | MG852011 |
| UFAH00042 | *Ilyonectria robusta* | 2017 | Tungurahua/Cevallos | - | MG852007 |

The extracted DNA was used for the amplification of a fragment of the histone 3 gene (*HIS3*), using primers CYLH3F, and CYLH3R [14], and a fragment of the translation elongation factor 1 $\alpha$ gene (*TEF*) using primers EF1 and EF2 [15]. PCR reactions contained 1X PCR buffer, 2 mM $MgCl_2$, 0.15 mM dNTP, 0.60 μM of each primer, 1.20 units of Taq DNA Polymerase, and 2.5 μL of DNA (40 ng/μL), in a final volume of 25 μL. Conditions for primers CYLH3F/ CYLH3R were 96 °C for 5 min, 30 cycles at 96 °C for 30 sec, 52 °C for 30 sec, and 72 °C for 1 min, and a final extension at 72 °C for 3 min. Conditions for primers EF1/EF2 were 94 °C for 10 min, 30 cycles at 94 °C for 1 min, 53 °C for 1 min, and 72 °C for 1 min, and a final extension at 72 °C for 10 min. PCR products were analyzed by agarose gel electrophoresis at 2%, stained with SYBR Green and visualized under ultraviolet light in a transilluminator.

The forward and reverse chains of each amplicon were sequenced by Macrogen Inc., South Korea and assembled using Geneious 6.0.6 (Biomatters Ltd.). The resulting contigs were used for species identification through BLASTn searches and later uploaded to GenBank. Sequences of the *HIS3* and the *TEF* of 13 representative taxa from *Ilyonectria* and *Dactylonectria* genera were downloaded from GenBank and used to perform phylogenetic analysis along with sequences generated on this study. Two taxa from the genus *Campylocarpon* were used as outgroup. Alignments and multilocus

phylogenetic analyses were carried out as described by Flores et al. [16]. A multiple sequence alignment was built for each fragment using the Muscle algorithm [17] in MEGA 7.0.21 [18]. Ambiguous regions were removed using Gblocks 0.91b [19]. The maximum-likelihood phylogenetic analysis (ML) was performed with RAxML 8.2.9 [20], in the CIPRES Science Gateway [21] and the Bayesian tree was built using BEAST 1.8.1., combining two runs of two million Markov chain Monte Carlo each with a 10% burn-in [22].

### 2.4. Pathogenicity Tests

Pathogenicity tests were performed in 35 six-month-old Andean blackberry plants, which were multiplied in vitro and transplanted to a sterile substrate (pumice and soil 1:3). Isolates from four different species of fungi (i.e., *Dactylonectria torresensis* (A. Cabral, Rego, and Crous) L. Lombard and Crous: UFAH00033, *Ilyonectria vredehoekensis* L. Lombard and Crous: UFAH00034, *Ilyonectria venezuelensis* A. Cabral and Crous: UFAH00041, and *Ilyonectria robusta* (A.A. Hildebr.) A. Cabral and Crous: UFAH00042) were independently inoculated into seven plants each. The inoculum suspension was prepared by filtering 15-day-old mycelium grown in potato dextrose agar (PDA) through a four-layered sterile gauze. The inoculum was adjusted to a final concentration of $1 \times 10^6$ conidia/mL with sterile distilled water [23]. Before inoculation, roots were disinfected with a 1% NaClO solution for 2 min and rinsed twice with sterile water. Roots were wounded at 2 cm from the apex, submerged in the inoculum for 30 min and transplanted to the substrate. Seven control plants were treated in the same way; however, they were immersed only in sterile distilled water. The experiment was done in a greenhouse at an average temperature of 20 °C and 62% relative humidity. Re-inoculation was carried out one month later using the same methodology.

The aggressiveness of the isolates was evaluated taking into account two variables: wilting of the basal leaves and necrosis of the crown. Root and leaf symptoms were evaluated fifteen weeks after inoculation, using scales of 0–4 for wilt in basal leaves: 0 = no symptoms; 1 = slight wilt, mild chlorosis; 2 = moderate wilt, discoloration in the leaves from the edges to the central rib, loss of turgidity and brown coloration; 3 = severe wilt, the edges of the leaves begin to curl and total loss of turgidity; 4 = dry leaves, the leaves show total discoloration, tend to peel off. Necrosis of the crown, 0 = no necrosis; 1 = 1%–25% necrosis with a violet coloration at the edges of the root; 2 = 26%–50% necrosis; 3 = 51%–75% necrosis; 4 = completely necrotic vascular bundles. Wilting of the basal leaves and infection of the crown was compared for each isolate with the non-parametric Kruskal–Wallis rank sum test. The medians with their corresponding 95% confidence intervals, obtained by non-parametric bootstrapping (10,000 bootstraps) with the rcompanion package, were plotted using the ggplot package. All statistical analyses were performed using R version 3.6.0 (R Development Core Team). Fungi were re-isolated from the roots of eight inoculated *R. glaucus* plants (two for each inoculated fungus) and identified using the protocols described for the original isolates.

## 3. Results

### 3.1. Isolates

After seven days of incubation at room temperature, tan to brown colonies were observed growing from the plant tissue in 90% of the samples. Monosporic isolates presented a concentric, radial growth of sparse and cottony mycelium, with brown and yellow pigmentation; showing white, brown, and beige edges. All colonies contained cylindrical, hyaline macroconidia with one to three septa and oval microconidia with one or no septum. Few globose, brown to yellowish chlamydospores forming in the hyphae or in macroconidia were also observed.

### 3.2. Morphological Characterization

#### 3.2.1. Dactylonectria torresensis

Culture characteristics: Mycelium irregular of brown coloration, cottony surface and even margin (Figure 2a). Conidiophores solitary arising laterally from aerial mycelium. Macroconidia predominating, 1–3 septate, straight, cylindrical, slightly broadly rounded (Figure 2b). 1-septate macroconidia, (20.73) 25.40–29.28–33.15 (36.79) × (3.48) 3.82–4.24–4.66 (5.06) μm; 2-septate macroconidia, (25.20) 26.43–29.94–33.45 (37.72) × (3.0) 3.38–3.92–4.45 (5.03) μm; and 3-septate macroconidia, (30.06) 33.08–37.13–41.17 (46.16) × (3.56) 3.96–4.44–4.92 (5.86) μm. Microconidia 0–1 septate, aseptate microconidia, (8.39) 9.68–11.18–12.67 (13.96) × (2.70) 3.20–3.65–4.09 (4.60) μm; 1-septate microconidia, (11.19) 12.96–15.61–18.26 (20.70) × (3.25) 3.71–4.40–5.10 (6.02) μm (Figure 2c).

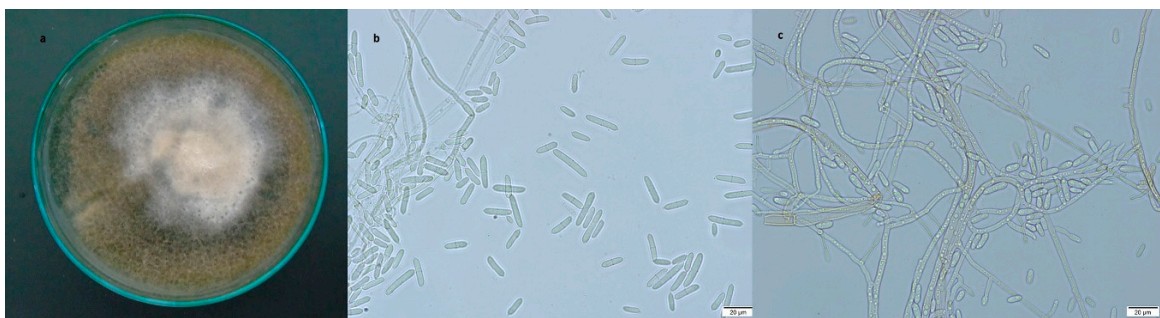

**Figure 2.** *Dactylonectria torresensis* (**a**) colony growing on potato dextrose agar (PDA), (**b**) macroconidia, and (**c**) simple conidiophore growing from the mycelium. Bars = 20 μm.

#### 3.2.2. Ilyonectria robusta

Culture characteristics: Mycelium cottony with moderate density, color slightly brown (Figure 3a). Simple conidiophores arising from the end of the aerial mycelium can have up to three cylindrical phialides slightly elongated and narrow at the top (Figure 3b). Macroconidia 1–3 septa with cylindrical shape and rounded ends. 1-septate macroconidia, (22.70) 25.09–28.36–31.63 (34.76) × (4.17) 4.99–6.12–7.25 (8.15) μm; 2-septate macroconidia, (25.61) 28.25–31.11–33.97 (37.86) × (4.78) 5.43–6.10–6.76 (7.21) μm; and 3-septate macroconidia, (30.78) 33.71–37.26–40.82 (43.36) × (4.45) 5.28–6.12–6.96 (8.01) μm. Aseptate microconidia, ovoid shape, (6.28) 8.14–10.02–11.89 (13.87) × (2.51) 2.87–3.35–3.82 (4.20) μm; 1-septate microconidia, (7.13) 10.95–13.80–16.64 (17.90) × (2.90) 3.27–3.87–4.47 (4.96) μm. Chlamidospores (Figure 3c) globes with thick wall and brown coloration.

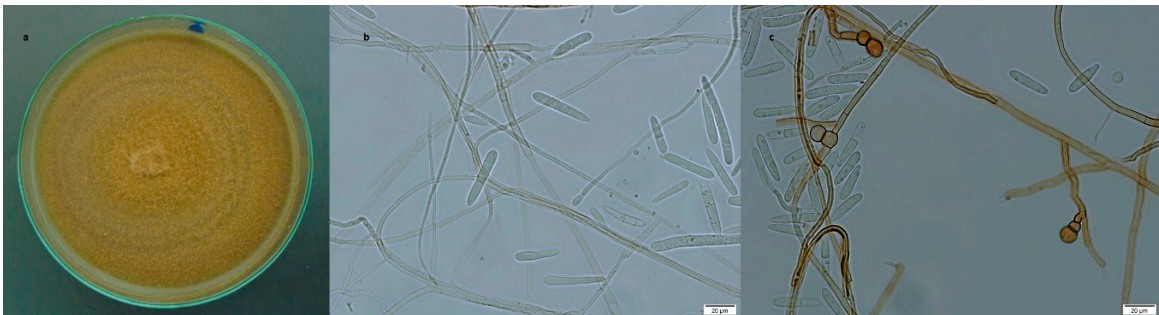

**Figure 3.** *Ilyonectria robusta* (**a**) colony growing on PDA, (**b**) simple conidiophores, and (**c**) chlamydospores in aerial mycelium. Bars = 20 μm.

#### 3.2.3. Ilyonectria venezuelensis

Culture characteristics: Mycelium growth irregular, cottony texture, brown coloration in the center and at the yellowish ends (Figure 4a). Conidiophores simple or complex (Figure 4b) arising

laterally from aerial mycelium, macroconidia predominate (Figure 4c) 1–3 cylindrical septa with rounded ends. 1-septate macroconidia, (22.05) 23.21–24.86–26.52 (28.62) × (3.72) 4.34–4.73–5.12 (5.43) µm; 2-septate macroconidia (26.55) 27.15–30.58–34.0 (38.38) × (4.81) 5.05–5.48–5.90 (6.39) µm; and 3-septate macroconidia (30.57) 31.93–34.02–36.11 (39.26) × (4.92) 5.33–5.84–6.35 (7.62) µm. Microconidia 0–1 septum, ellipsoidal or ovoid. Aseptate macroconidia (6.21) 7.80–8.87–9.94 (11.02) × (1.88) 2.17–2.61–3.04 (3.73) µm; 1-septate microconidia (12.10) 12.59–14.33–16.07 (19.88) µm.

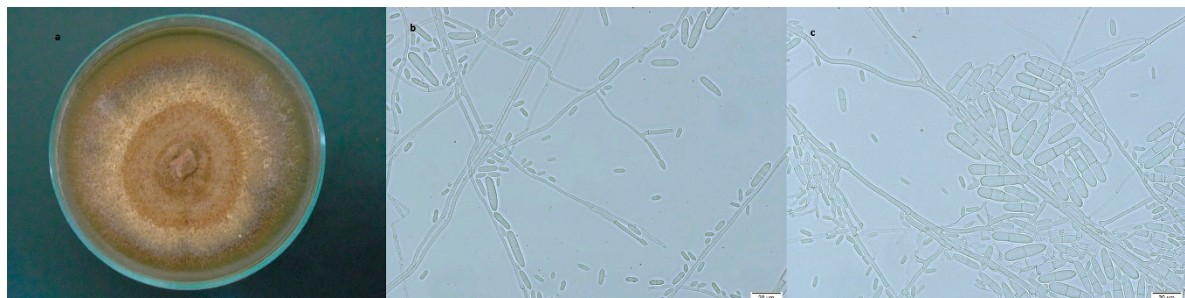

**Figure 4.** *Ilyonectria venezuelensis* (**a**) colony growing on PDA, (**b**) simple conidiophores and 1-septate macroconidium, (**c**) micro- and macroconidia. Bars = 20 µm.

### 3.2.4. *Ilyonectria vredehoekensis*

Culture characteristics: Mycelium with irregular branched growth, dark brown cottony texture and small woolly creamy clusters at the ends (Figure 5a). Conidiophores that arise from the aerial mycelium (Figure 5b). Macroconidia of 1–3 septa predominate, cylindrical shape with rounded ends. 1-septate macroconidia, (22.54) 26.26–23.81–28.80 (32.08) × (4.07) 4.41–5.02–5.63 (6.70) µm; 2-septate macroconidia, (24.44) 26.41–28.10–29.78 (30.94) µm; and 3-septate macroconidia, (30.26) 31.70–33.90–36.10 × (5.16) 5.48–5.96–6.44 (7.16) µm. Microconidia without septum were oval globose to fusiform, (3.75) 5.34–6.79–8.25 (9.62) µm; 1-septate microconidia, (12.50) 13.83–16.06–18.29 (19.70) × (3.25) 3.70–4.40–5.09 (6.02) µm. Chlamydospores globose shaped, thick walled, brown (Figure 5c).

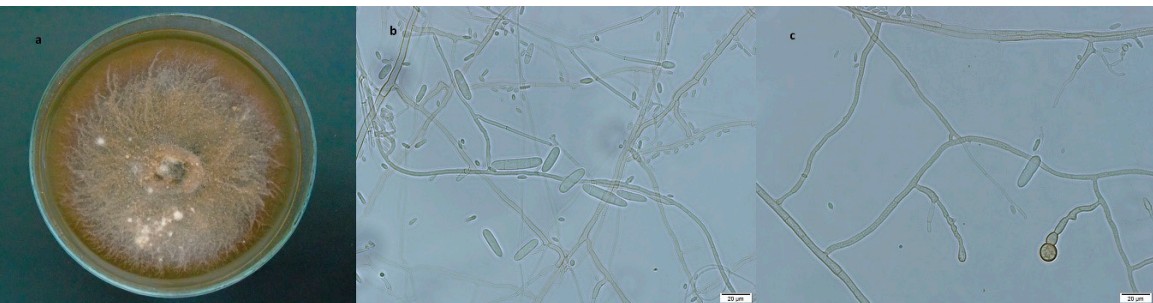

**Figure 5.** *Ilyonectria vredehoekensis* (**a**) colony growing on PDA, (**b**) micro- and macroconidia, (**c**) chlamydospores growing from mycelium. Bars = 20 µm.

### 3.3. Molecular Identification and Phylogenetic Analysis

Amplicons of 500 and 750–850 bp were obtained for the *HIS3* and *TEF*, respectively. Sequences of the two genes indicated that isolates belonged to each of seven different species, namely *Dactylonectria novozelandica* (A. Cabral and Crous) L. Lombard and Crous, or *D. torresensis*, *Ilyonectria europaea* A. Cabral, Rego, and Crous, *I. robusta*, *I. venezuelensis*, *I. vredehoekensis*. The ML and Bayesian two-loci phylogenetic trees were congruent, showing two clades corresponding to the *Dactylonectria macrodidyma* and the *I. radicicola* complex, respectively (Figure 6). Isolates corresponding to the *D. macrodidyma* complex, including *D. torresensis* and *D. novozelandica* were located only in the province of Tungurahua. Isolates of the *I. radicicola* complex were found in Bolivar and Tungurahua, with *I. vredehoekensis* being the most common species with eight isolates, followed by *I. venezuelensis* with two isolates, and

*I. robusta* and *I. europaea* with one isolate each. Isolate UFAH00038 did not group with any of the species described so far in the literature and it may represent a new species of *Ilyonectria*.

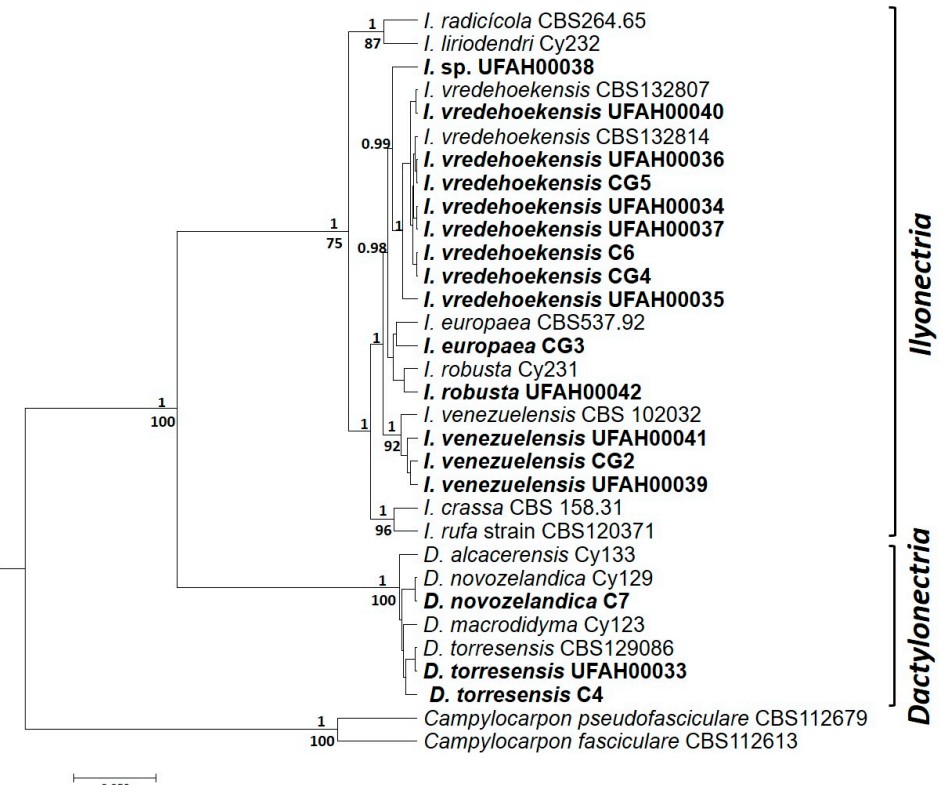

**Figure 6.** Bayesian multilocus tree of *Ilyonectria* and *Dactylonectria* genera using the *HIS3* and *TEF* loci. The tree was built on BEAST 1.8.1 combining two runs of two million Markov chain Monte Carlo each. Values above the nodes represent posterior probabilities and values below the nodes represent bootstrap support from a maximum-likelihood analysis performed in RaxML.

### 3.4. Pathogenicity Tests

Typical symptoms of black foot disease were observed on infected plants fifteen weeks after inoculation. The vasculature of the root and crown was necrotic, showing clear and dark brown pigmentation that contrasted with the healthy tissue from control plants. Some of the basal leaves died and others wilted, showing chlorosis, necrosis of the edges of the leaf, and curling (Figure 7a). Non-inoculated plants were asymptomatic (Figure 7b). Isolates recovered from twelve symptomatic plants (three plants for each fungus species used as inoculum) were identified morphologically and molecularly as *D. torresensis*, *I. robusta*, *I. venezuelensis*, or *I. vredehoekensis*; confirming that these species can cause black foot in Andean blackberry plants.

While plants inoculated with different species of *Ilyonectria* or *Dactylonectria* were symptomatic, showing moderate to severe wilt, none of the non-inoculated control plants showed symptoms (Figure 7). The lower leaves of most of the inoculated plants displayed chlorosis, necrosis advancing from the edges to the central vein, and loss of turgidity. Violet necrotic lesions were observed in the vasculature of the crown of inoculated plants. No symptoms were observed on the crown of the controls. Even though the severity of black foot caused by *I. vredehoekensis* was the highest among the four fungal species tested, the differences were not significant in either response variable (Figure 8). Morphological and molecular identification of the fungi that were re-isolated from symptomatic tissue after the pathogenicity tests confirmed that each plant was infected by the fungus that it was originally inoculated with, completing Koch's postulates. There was no fungal growth in plates with plant tissue from non-inoculated controls.

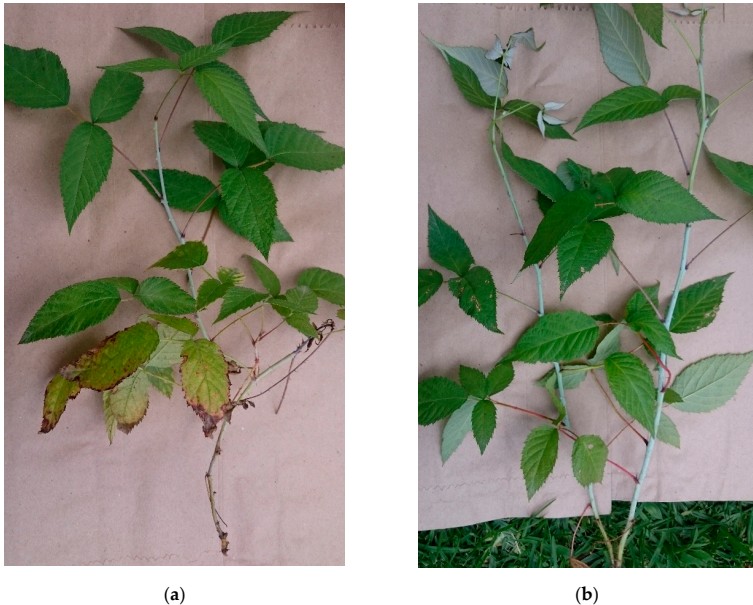

(**a**)            (**b**)

**Figure 7.** (**a**) *Rubus glaucus* plant at fifteen weeks after inoculation with *Ilyonectria vredehoekensis*, (**b**) non-inoculated control.

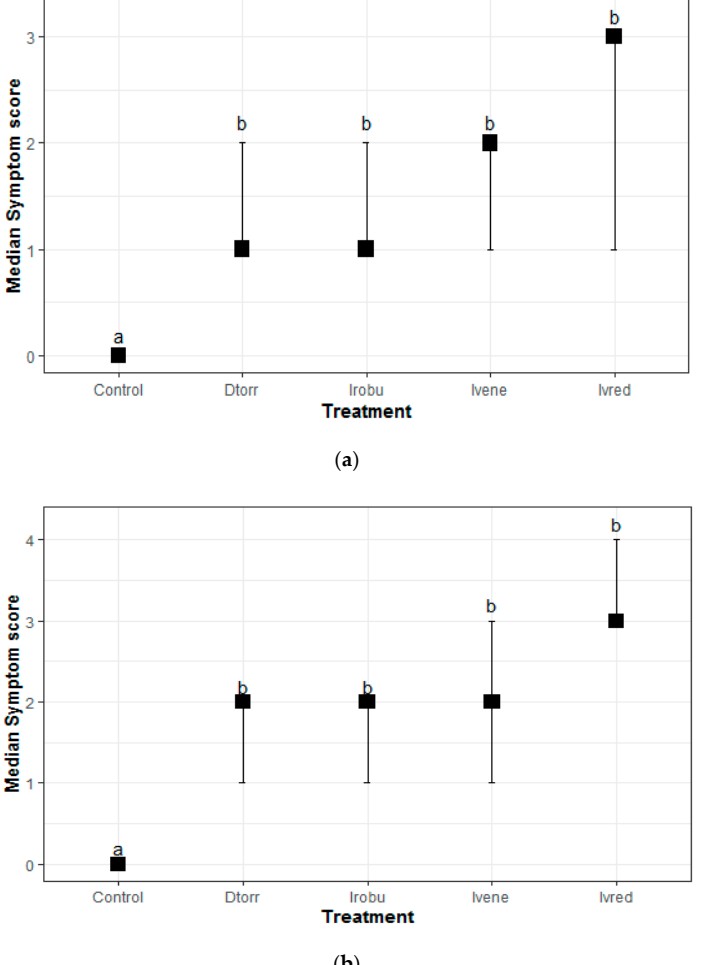

**Figure 8.** Results of the pathogenicity tests of *D. torresensis*, *I. robusta*, *I. venezuelensis*, and *I. vredehoekensis* on *Rubus glaucus*. Two variables were measured fifteen weeks after inoculation, (**a**) wilting in the basal leaves and (**b**) necrosis of the crown.

## 4. Discussion

Black foot disease of vines has been reported in several countries, being associated mainly with species of the genera *Campylocarpon*, *Dactylonectria*, *Ilyonectria*, and *Neonectria*. These fungi have been associated with black foot disease in *Protea* spp., in South Africa [9], and in *Vitis vinifera* L. with a wide geographical distribution that includes Australia, Canada, New Zealand, Portugal, South Africa, Spain, and the United States [10,23,24]. In Italy, it was also reported that *D. torresensis* can cause root rot of *Viburnum tinus* L. [25]. In our study, from 2014 to 2017, we recovered isolates representing six species of *Dactylonectria* and *Ilyonectria* from symptomatic Andean blackberry plants. Even though we were able to characterize isolates belonging to all six species at a molecular level, we were only able to do the morphological characterization and pathogenicity tests in four species due to the loss of *I. europaea* and *D. novozelandica* isolates, which were collected in 2014 and were contaminated during storage.

Asexual spores were observed in all the isolates that were characterized morphologically; however, sexual structures were not found. Morphological identification of species with *Cylindrocarpon*-like anamorphs is difficult, making DNA sequencing a necessity when working with this group of fungi [13]. In this work, fragments of the *TEF* and *HIS3* genes were amplified and sequenced for the identification of isolates at the species level. Both molecular markers have been used in previous studies for the identification of species within the *I. radicicola* and *D. macrodidyma* complexes as their sequences contain unique polymorphisms for each species [10].

Pathogenicity tests showed that the *D. torresensis*, *I. robusta*, *I. venezuelensis*, and *I. vredehoekensis* reached the vasculature of the crown, entering through the root system as previously reported in blackberry and grapevines [4,7,23], causing necrosis of the vascular bundles and wilting. According to Kluge [26], the violet tone observed in the vasculature of the crown is given by a substance of phenolic nature secreted by this group of pathogenic fungus. On the other hand, Cedeño et al. [7] established that the pathogenicity of *C. destructans* is related to the production of a toxin that weakens and kills the root tissue of plants affected by stress, transplantation, and pruning at the root level. The previous report of black foot disease in Andean blackberry [7] identified *C. destructans* var. *destructans* as the causal agent, but this identification was done based only on morphology, before sequence-based taxonomy of *Cylindrocarpon*-like fungi was in place. According to our knowledge, this is the first report of *Ilyonectria* and *Dactylonectria* species associated with black foot in Andean blackberry, a disease with a 13.3% incidence in the main production zones of Ecuador.

The size and septation of macroconidia and microconidia of *D. torresensis* isolated from Andean blackberry agree within the description by Cabral et al. [10] and the shape of its conidiophores (elongated, with a slight widening and minute curves) is in accordance with the observations by Weber and Entrop when they determined the association of this fungus with black root rot of strawberry and blackberry [6]. The morphology of the *I. robusta* and *I. venezuelensis* isolates described in our study is congruent with the definitions by Cabral et al. for each of those species [8]. Likewise, the description of our *I. vredehoekensis* isolate complies with the definition by Lombard et al. [9].

## 5. Conclusions

Different species of *Ilyonectria* and *Dactylonectria* can cause black foot disease of Andean Blackberry, producing symptoms of wilting and root tissue necrosis. The fungi enter through the root system, colonizing the vasculature of the stem where tissue becomes necrotic, resulting in symptoms of severe wilting. Even though *I. vredehoekensis* showed the greatest disease severity among the four species that were analyzed, the differences between fungi were not significant for any of the variables analyzed.

Even though the phenotypic characteristics observed in this study conform to what is described in the literature, sequence analysis is fundamental for the specific identification of this group of fungi which was earlier treated as a species complex. The molecular and morphological characterization of fungi causing black foot in Andean blackberry will allow the implementation of adequate control measures for the disease.

**Author Contributions:** Conceptualization, F.F., W.V., and L.A.; methodology, J.S., P.I., D.M., A.M., and N.P.; investigation, J.S., P.I., D.M., and N.P.; formal analysis, P.I., J.S., and F.F.; writing (original draft preparation and review) and editing, J.S., P.I., C.T., and F.F.; supervision, F.F., A.K., L.A., and W.V.; project administration, W.V., A.K., and C.T.; resources, A.K., W.V., A.M., and C.T.

**Funding:** Authors thank the New Zealand Ministry of Foreign Affairs and Trade for funding this research through the project "Biocontrol for Sustainable Farming Systems—Ecuador".

**Acknowledgments:** All samples were collected and sequenced under framework contract MAE–DNB–CM–2017–0071.

**Conflicts of Interest:** The authors declare no conflict of interest.

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
