# Peer review of "Dactylonectria and Ilyonectria Species Causing Black Foot Disease of Andean Blackberry (Rubus Glaucus Benth) in Ecuador"

_diversity, doi:10.3390/d11110218_

Round 1

Reviewer 1 Report

Dear authors,

This work is interesting and provides the identification of the causal agents of black foot disease of Andean blackberry, including a morphological and molecular characterization of the species.

The methods and results are described clearly, arranged logically and are easy to follow. The data is adequate and the results are presented clearly. I would suggest to improve the background and the conclusions.

In order to follow the taxonomic rules, the name of the authors should be included after the name of the species at least once in the text.

About material and methods, in the fungal isolation, did you not use any supplements in the media to minimise the growth of bacteria or to obtain more compact fungal colonies?

You have provided a morphological characterization of the fungal species. Have you compared these characters with ones in the original description of the species? Are your morphological characterization and the specie description in accordance? Would be better include this information in the results.

Keywords: could be changed to highlighted better the issue

Please check for further typos and I wish you good luck with your manuscript and your ongoing research.

Introduction:

Line 40 – not italic after Benth).

Material and methods

Line 79 – what was the concentration of the fungal DNA in the PCR reactions?

Line 135 - correct Table number 2 with Table 1

Line 163 – correct Clamidospore with Chlamidospore

Results

Line 250 – correct D novozelandica with D. novozelandica

Line 315 and 136 – space between lines

Author Response

Dear Reviewer,

Thanks for taking the time to revise our manuscript and for your valuable suggestions. We have attempted to address all your comments as follows:

C: The methods and results are described clearly, arranged logically and are easy to follow. The data is adequate and the results are presented clearly. I would suggest to improve the background and the conclusions.

A: The introduction and conclusions were extended including more important background information

C: In order to follow the taxonomic rules, the name of the authors should be included after the name of the species at least once in the text.

A: The authorities have been included after all species current scientific names the first time they appear in the text.

C: About material and methods, in the fungal isolation, did you not use any supplements in the media to minimise the growth of bacteria or to obtain more compact fungal colonies?

A: We revised the methods and, in fact, PDA was amended with 100 ppm chloramphenicol for the fungal isolations. This is now included in line 82

C: You have provided a morphological characterization of the fungal species. Have you compared these characters with ones in the original description of the species? Are your morphological characterization and the specie description in accordance? Would be better include this information in the results.

A: The comparison with the original descriptions is now included in lines 336-342

C: Keywords: could be changed to highlighted better the issue

A: New keywords are now used: Cylindrocarpon-like anamorphs, root rot, soilborne pathogens, wilt

C: Please check for further typos and I wish you good luck with your manuscript and your ongoing research.

A: Thank you!

Introduction:

C: Line 40 – not italic after Benth).

A: Fixed in line 43

Material and methods

C: Line 79 – what was the concentration of the fungal DNA in the PCR reactions?

A: Concentration included in line 110

C: Line 135 - correct Table number 2 with Table 1

A: Fixed in line 172

C: Line 163 – correct Clamidospore with Chlamidospore

A: Fixed in line 201

Results

C: Line 250 – correct D novozelandica with Dnovozelandica

A: Fixed in line 312

C: Line 315 and 136 – space between lines

A: Fixed, space in line 422

Please see attached file with the corrections

Best Regards

Francisco

Reviewer 2 Report

Dear authors

the study is concrete and very nice to show important pathosystem.

please fix something and make much better the introduction which is very poor at the moment.

A few suggests are present into the attached file.

Author Response

Dear Reviewer,

Thanks for taking the time to revise our manuscript and for your valuable suggestions. We have attempted to address all your comments as follows:

C: Please fix something and make much better the introduction which is very poor at the moment.

A: The introduction and conclusions were revised and extended. These sections now include more relevant information

C: A few suggests are present into the attached file.

A: All suggested changes are now included in the manuscript. 

Please see attached file for the final version.

Best Regards

Francisco
